# Hydrophobization of Monolithic Resorcinol-Formaldehyde Xerogels by Means of Silylation

**DOI:** 10.3390/gels8050304

**Published:** 2022-05-16

**Authors:** Fabian Henn, René Tannert

**Affiliations:** Institute for Materials Research, Departments of Aerogels and Aerogel Composites, German Aerospace Center (DLR), Linder Höhe, 51147 Köln, Germany; fabian.henn@dlr.de

**Keywords:** xerogels, resorcinol-formaldehyde, functionalization, hydrophobicity

## Abstract

In materials research, the control of wettability is important for many applications. Since they are typically based on phenolics, organic aerogels, and xerogels are intrinsically hydrophilic in nature, and examples of the chemical functionalization of such gels are scarce and often limited to powders. This study reports on the silylation of monolithic resorcinol-formaldehyde (RF) xerogels using solutions of silyl chlorides and triflates, respectively, in combination with an amine base. The resulting gels are structurally characterized by means of elemental analysis, X-ray photoelectron spectroscopy, pycnometry, sorption analysis, and scanning electron microscopy with electron-dispersive X-ray spectroscopy. The wetting behavior of the silylated gels was studied by the determination of the contact angle to water after exposure of the gels to ambient air. Additionally, the uptake of liquid water and aqueous acids and bases was investigated. As a result, processes for the functionalization of RF xerogels with sterically demanding silyl moieties have been established. Although the analyses indicate that silylation occurred to a rather small extent, highly hydrophobic gels resulted which retained the wetting behavior over the course of several months with contact angles of >130°. Monoliths bearing sterically demanding silyl groups showed higher stability towards aqueous acid than trimethylsilylated RF gels.

## 1. Introduction

Understanding wetting phenomena and ultimately producing a surface with controlled wettability remains a challenging target of material science. Such endeavors include the chemical functionalization of a material’s surface [1,2,3,4,5,6,7,8].

Aerogels and xerogels based on resorcinol-formaldehyde (RF) polymers have attracted widespread interest due to their high porosity and low density, resulting in properties that render them attractive insulating materials for a wide range of applications [9].

However, the free phenolic groups of the polymer render the material rather hydrophilic in nature, thus limiting the scope of potential applications to some extent. In order to overcome this drawback, the phenolic hydroxy groups can either be circumvented by using functionalized precursors for polymerization [10], or by functionalization in a post-synthetic fashion (Figure 1). Prior efforts for post-synthetic hydrophobization of RF include alkylation: Alonso-Buenaposada et al. conducted gas-phase methylation of RF xerogel powders, resulting in a material that proved superhydrophobic (static contact angle of water >150°) for a period of 40 days. However, the methodology was only demonstrated in powdered material, and the synthesis involved rather harsh conditions (methanol at 240 °C) [11].

Recently, Lermontov et al. achieved fluoroacylation of monolithic RF xerogels using ethereal solutions to the corresponding fluorinated anhydrides. When long fluoroalkyl chains were introduced, significant hydrophobicity (water contact angles up to 145°) was observed. However, no details on the longevity of the hydrophobic effect had been reported [12].

Another approach toward hydrophobic RF gels is their silylation. Silyl reagents are commonly employed in solution phase synthetic organic chemistry for the (temporary) chemical protection of otherwise reactive functional groups, including phenols [13]. For the same purpose, silylation has also been performed in solid-phase organic synthesis [14]. These efforts, however, typically aim at the temporary masking of the reactive hydroxy group, i.e., reducing its Brønsted acidity. The use of silylation in materials chemistry for the purpose of hydrophobization of porous material is less common and basically restricted to inorganic oxides (silica foremost) treated with trimethylsilyl chloride vapor affording trimethylsilyl ethers. The commercially relevant Cabot process relies on silicon tetrachloride and alcohols or hexaalkyl siloxanes for the production of alkylated silica via alkylsilyl chlorides formed in situ. Apart from inorganic samples, the trimethylsilylation of cellulose has been described, both in solution and in the gas phase, using disilazanes [15].

As most silylations of monolithic gels have been reported in the gas phase, these reactions have thus far only been reported with small silyl reagents and in the absence of bases. While small silyl reagents (such as trimethyl silyl chloride) often show higher reactivity than bulkier reagents, the hydrolysis of the resulting silyl ether is also favored, resulting in short-term hydrophobicity of such material. This finding can be easily rationalized by the steric accessibility of the electrophilic silicon atom: bulky groups, such as *tert.*-butyl dimethylsilyl (TBDMS) or triisopropylsilyl (TIPS) can withstand nucleophilic attack more efficiently. On the other hand, electronic effects can render more elaborate silyl moieties much more resistant to acidic and/or alkaline media [16,17]. Furthermore, examples of the employment of more elaborate silyl reagents in the functionalization of porous bodies can be found for inorganic systems, such as clays [18,19,20,21,22].

One of the few examples of silylations of RF xerogels in solution was reported in a patent by E. I. du Pont de Nemours and Company. There, a number of different silyl reagents, including disilazanes and chlorides, were investigated. Hydrophobic effects and the shelf-life of the resulting gels were not described though [23].

Alonso-Buenaposada et al. used hexamethyldisilazane as a reagent that bears reactive trimethylsilyl groups and ammonia as a base generated in situ. Monolithic RF gel was ground to the powder prior to treatment with the neat disilazane at 80 °C. Superhydrophobic effects were reported to last for about three weeks at 20 °C and 80% relative humidity [24].

In the solution phase, organic synthesis *O*-silylations of phenols are typically conducted using silyl chlorides and amine bases in polar-aprotic solvents [25]. When the reactivity of silyl chlorides is too low due to increasing steric hindrance, one solution is to release tight ion pair between the silyl cation and chloride counterion using less nucleophilic and therefore, less coordinating anions (e.g., triflates) and bases (e.g., 2,6-lutidine) [26].

In order to achieve long-lasting hydrophobicity for RF xerogels as well as stability against aqueous acids and bases, we have investigated the synthetic routes for the silylation of monolithic RF xerogels in solution using external bases. The reagents employed also include sterically demanding silyl reagents and electronically activated silyl triflates [27].

## 2. Results and Discussion

Using conditions reported by Schwan et al., the synthesis of Fricke-type RF xerogels, allowing for ambient drying of the monoliths, was performed [28,29].

Silylation experiments were performed with a variation of the silyl precursors and amine bases. Polar-aprotic solvent (DMF) was used throughout the experiments. The successful incorporation of silicon could be established through elemental analysis. A summary of the elemental compositions is displayed in Table 1. A general notion is that silicon is present in quantities of <2% by weight, regardless of the source of silicon. In view of the theoretical Si contents of up to 20% (m/m), assuming silylation of all aromatic hydroxy groups present in the RF xerogel, incomplete conversion of hydroxyl groups can be deduced. The assumption that free hydroxy groups are still present in the gels is supported by rather broad bands centered at around 3200 cm^–1^ in the infrared spectrum (data not shown). As the employment of more reactive reagents for silylation (triflate and lutidine) did not alter the silicon content much, the *chemical reactivity* of the phenolic hydroxy groups in the gel seems not to be crucial, but rather their *accessibility* within the pore network.

In order to investigate the chemical nature of the silicon within the monoliths, X-ray photoelectron spectroscopy (XPS) was performed (Figure 1). The resulting bond energies (for O1s orbital: Figure 1c; for Si–2p orbital: Figure 1d) clearly indicate the presence of silicon-oxygen bonds supporting the formation of a covalent silyl ether [15].

Further support for the covalent coupling of the silylating agents is provided by the XPS signals in the Si 2p region: a deconvolution of the experimentally observed signal and its fitted envelope function suggests a ratio of Si–O bonds to Si–C bonds that correspond to the expected 1:3 ratio for trialkylsilyl ethers. In addition, Fourier-Transform Infrared (FT-IR) studies performed with TBS-RF revealed a new band at 839 cm^−1^, corresponding to Si–O stretching vibrations (data not shown).

Scanning electron microscopy (SEM) was performed on the RF gel precursor (Figure 2a) as well as silylated gel TBS-A (Figure 2b). Both microstructures contain interconnected spherical particles with diameters in the single-digit µm range and large pores in between the backbone. Therefore, it can thus be concluded that the treatment with silylating reagents did not significantly affect the microstructure, nor the porous network within the xerogel.

Energy-dispersive X-ray spectroscopy (EDX) was applied, along with scanning electron microscopy. Here, a rather homogeneous distribution of carbon, oxygen, and silicon, among others, was observed (Figure 3). The lower relative abundance of Si agrees with the hypothesis that not all phenolic hydroxy groups are silylated.

Krypton sorption experiments, followed by BET analysis, gave rise to the inner surface area. Here, values were virtually unchanged when comparing RF to its silylated pendants. Moreover, the porosity, as determined by pycnometry, was only slightly reduced by silylation; thus underlining the assumption that the pore network is not affected by the functionalization process (Table 2).

In order to explore the hydrophobicity of silylated RF xerogels, static contact angles were determined by means of drop shape analysis after the exposure of monoliths to ambient air and the subsequent placement of water droplets onto the monoliths’ surface. Here, both untreated RF and TMS-functionalized gels with low silicon content (TMS-A) rapidly absorbed water droplets, preventing the determination of a contact angle and emphasizing the hydrophilic nature of the gels (Figure 4a).

On the other hand, gels with higher TMS contents (TMS-B) or those bearing sterically more demanding silyl groups (TBS, TIPS, TBDPS) did bear water droplets, thus allowing for quantitative analysis of their hydrophobic properties (Figure 4b).

When the static contact angle was determined by the manual positioning of the tangent between the surface and the droplet (tangent method), a range of contact angles of 133.9–143.9° resulted (Table 3). For comparison, an automatized positioning of the tangent was also employed, using the low-bond axisymmetric drop shape analysis (LB-ADSA) method [31]. As a result, consistently higher values (137.1–151.1°) were obtained. However, by using this approach, the tangents were placed in a symmetrical fashion on both sides of the droplets using the LB-ADSA approach, while significant deformations of the droplets were observed in our experiments. Therefore, that method was discarded and replaced by the manual positioning of the tangents within the ImageJ software (tangent method).

In order to determine the wetting behavior for the whole monolith rather than individual spots, dynamic contact angles were also determined by applying a modified version of the Wilhelmy plate method (see above). As a result, contact angles of 101.8–152.4° were determined (Table 4).

Therefore, in comparison to the static contact angles, dynamic values were in a similar range, albeit with a higher experimental deviation. As the whole monolith is submerged into the water, the lower accuracy of the plate method may be attributed to the roughness that is inherently high for such xerogel monoliths. Deviations may also stem from local inhomogeneities on the surface.

The significant hydrophobicity of most silylated xerogels is remarkable, given their low silicon content. A possible explanation includes the presence of more accessible surfaces that are more readily functionalized (see above) and—when silylated—contribute to a larger degree to the hydrophobicity of the monolith.

While a silicon content of 0.61% in TMS-A is not sufficient to render gel monoliths bearing trimethylsilyl groups hydrophobic, a mere 0.27% of silicon is sufficient in the case of bulky triisopropylsilyl groups (TIPS-B). Therefore, the steric nature of the silyl moiety seems to be at least as crucial to hydrophobicity as is the actual silicon content.

The behavior of the monoliths within liquid water was studied by the measurement of their gain in mass upon their submersion in water. While the TMS-functionalized sample gained even more weight than the RF reference, samples functionalized with TBS or TIPS took up significantly less water and samples, and the effect was still pronounced after a period of >7 weeks (Figure 5a). Most likely, the trimethylsilyl group was hydrolyzed, leading to a rapid loss in hydrophobicity, while the bulkier silyl groups were more inert towards hydrolysis. These findings are in accordance with the literature on silyl protective groups in the organic synthesis, where TMS groups are often too labile for the effective protection of phenols [13].

The silylated xerogel monoliths were also assessed for their stability toward acids and bases. Treatment with 10% HCl in water led to TMS gels taking up water in the range of unfunctionalized RF, while TBS- and TIPS-modified gels also took up significantly less water when treated for several weeks (Figure 5b). Again, the higher steric demand of the tertiary butyl and isopropyl groups in TBS and TIPS, respectively, account for the increase in inertness towards acidic hydrolysis. In contrast, the treatment with 10% NaOH in water resulted in a reduced uptake of liquid for TBS- and TIPS-functionalized samples, but the effect did not persist over several days (Figure 5c).

## 3. Conclusions

In summary, a procedure for the silylation of monolithic resorcinol-formaldehyde xerogels has been established. The procedure involves a variety of sterically encumbered silyl reagents and amine bases, including electronically activated triflates.

Gels silylated in such a manner show a marked hydrophobicity with contact angles consistently exceeding 130°. The hydrophobic characteristics of the monoliths are retained when the gels are exposed to air humidity for several months. While gels functionalized with small TMS groups readily hydrolyze in liquid water, sterically demanding silyl groups render RF xerogel monoliths resistant to water and even diluted hydrochloric acid for weeks, while they tend to be less resistant towards the dilute base.

## 4. Materials and Methods

Resorcinol (R, 98% purity, Sigma Aldrich) was purchased from VWR International GmbH, Darmstadt, Germany. The aqueous solution of formaldehyde (24% m/m, stabilized with 0.5–1.5% methanol) was supplied from VWR International GmbH, Darmstadt, Germany. Anhydrous sodium carbonate (C) 99.6% was purchased from Arcos Organics. Nitric acid (32% m/m, Bernd Kraft) supplied by VWR International GmbH, Darmstadt, Germany, was diluted to a final concentration of 2.0 M using deionized water (W) obtained by passing tap water through a water purification cartridge (DIA 6000 MB, EnviroFALK GmbH, Westerburg, Germany). The silylating agents *tert*.-butyldimethylsilyl chloride (TBSCl, 97%, Sigma Aldrich), *tert*.-butyldiphenylsilyl chloride (TPBDPSCl, 98%, Sigma Aldrich), trimethylsilyl chloride (TMSCl, 98%, Sigma Aldrich), triisopropylsilyl trifluoromethanesulfonate (TIPSOTf, 97%, Alfa Aesar), triisopropylsilyl chloride (TIPSCl, 97%, ChemPUR), and trimethylsilyl trifluoromethanesulfonate (TMSOTf, 98%, ChemPUR) were purchased from VWR International GmbH, Darmstadt, Germany. The amine bases imidazole (≥99.5%, Sigma Aldrich) and 2,6-lutidine (redistilled, >99%, Sigma Aldrich) were acquired from Th. Geyer GmbH & Co. KG, Lohmar, Germany. The reagents and solvents were used without further purification.

As organic solvents, *N*,*N*-dimethylformamide (DMF, 99.5%, Baker, delivered by VWR International GmbH, Darmstadt, Germany) and acetone (pure, technical grade, Th. Geyer GmbH & Co. KG, Lohmar, Germany) were used. Molecular sieves (3 Å, 2.5 mm, 0.08–0.20 in, Alfa Aesar) procured from Th. Geyer GmbH & Co. KG, Lohmar, Germany, were activated before use by heating for 60 min at 200 °C, followed by treatment at 50 mbar and 200 °C for a period of >12 h.

The pH value was controlled using a SevenEasy pH instrument equipped with an InLab^®^ Expert pro pH electrode (relative accuracy of ±0.01, Mettler Toledo, Gießen, Germany). The residual water content within the solvents was determined by volumetric Karl Fischer titration using a TitroLine 7500 apparatus (SI Analytics, Mainz, Germany). Titrations were performed using CombiNORM 5K and Karl Fischer Solvent K, respectively (purchased from VWR International GmbH, Darmstadt, Germany).

For the determination of the sample mass, the fine balance of the New Classic MF ML 104 (Mettler Toledo GmbH, Gießen, Germany) was used. Sealable containers for gelation (125 mL, polypropylene (PP) with screw-cap) were acquired by VWR International GmbH, Darmstadt, Germany.

### 4.1. Synthesis of Xerogels

For the synthesis of resorcinol-formaldehyde (RF) xerogels, resorcinol (R), formaldehyde (F), deionized water (W), and sodium carbonate (C) were used as starting materials. The following molar ratios were chosen: R/W = 0.044; R/F = 0.74; and R/C = 1500. The following amounts, which are based on a single monolith, were applied to a total of 50 gel monoliths.

A glass beaker was charged with a cross-shaped magnetic stirring bar before the addition of solid resorcinol (0.027 mol, 3.0 g) and deionized water (0.0012 mol, 7.67 mL). After stirring (250 rpm) at 22 °C for 5 min using a magnetic stir-plate, a colorless solution resulted. Aqueous formaldehyde solution (24% m/m, 0.037 mol 4.35 mL) was added in one portion and the resulting solution was stirred at 22 °C for 5 min. Solid anhydrous sodium carbonate (0.018 mmol, 1.93 mg) was added in one portion. Stirring for another 5 min resulted in a colorless solution whose pH was adjusted to 5.5 by dropwise addition of nitric acid (2 N)**.** The solution was stirred at 22 °C for another 60 min before it was transferred to cylindrical polypropylene molds (per mold ca. 30 mL). The molds were closed, transferred into an oven (80 °C), and left for gelation and aging for a period of 2 days. The resulting red aquogels were removed from the molds and washed by treating the gels for 10–12 h with DMF, followed by draining (4 × 20 mL) the containing molecular sieves (3 Å). When the water content was <0.1%, as determined by coulometric Karl Fischer titration, the samples were subjected to silylation.

### 4.2. Silylation of Xerogels

The syntheses were conducted using either a combination of silyl chloride and imidazole as a base (method A) or using a silyl triflate in combination with lutidine as a base (method B).

A 100 mL-glass beaker was sequentially filled with solid imidazole (method A, 4 eq., 0.20 mol, 13.62 g) or liquid 2,6-Lutidine (method B, 4 eq., 0.20 mol, 23.22 mL) and solvent (DMF, 22.5 mL). Ultrasonication for a period of 5 min at 22 °C resulted in a colorless solution. Polypropylene molds were charged with xerogel monolith, and the solution of the amine base was added before the addition of the silylating agent (2.0 eq. 0.10 mol). While solid TBSCl (0.10 mol, 15.07 g) was added in one portion under air atmosphere, the more moisture-sensitive liquid silylating agents TBDPSCl (25.64 mL, 0.10 mol), TIPSOTf (26.88 mL, 0.10 mol), TIPSCl (0.10 mol, 21.23 mL) and TMSOTf (0.10 mol, 18.10 mL) were transferred in two portions under argon atmosphere using a 20 mL-syringe, and PP molds were sealed using Parafilm.

Monoliths were washed using DMF followed by acetone (each solvent: four iterations of addition, leaving for 12 h and draining) and dried in the fume hood (2 days at 20 °C) and an oven (3 days at 60 °C).

Plane-parallel surfaces were produced by treating the monoliths with abrasive paper (P120, Format, Germany). Residual dust was removed by applying a gentle air flux to the monoliths.

### 4.3. Characterization

Both the skeletal and envelope density of the xerogels were analyzed using a helium pycnometer, AccuPyc II 1340, and a sand pycnometer, GeoPyc 1360, respectively, both from Micromeritics GmbH, Unterschleissheim, Germany. For the former method, helium 5.0 from Linde GmbH, Pullach, Germany, was used. For the latter, DryFlo™, a fine-grained sand with carbon additive (purchased from Micromeritics, Unterschleissheim, Germany), was used and a test load of 51 N was applied. An AUW220 analytical balance (measurement accuracy: ±0.1 mg) from Shimadzu Europa GmbH, Duisburg, Germany, was used to weigh the specimens.

The internal surface area was determined by multipoint surface adsorption using a 3Flex Physisorption device (Micromeritics GmbH, Unterschleissheim, Germany) and evaluated according to Brunauer, Emmett, and Teller [32]. Krypton 5.0 from Linde GmbH, Pullach, Germany, was used as the adsorptive, and relative pressures p/p° of 0.05 to 0.30 were considered according to the IUPAC recommendations [33]. Prior to sorption analysis, samples were degassed for 3 h at 120 °C under reduced pressure (<10^−3^ mbar) before measurement using a Smart Vacprep apparatus (Micromeritics, Germany).

The X-ray photoelectron spectroscopic (XPS) analyses were performed by the Interdisciplinary Center for Analytics on the Nanoscale (ICAN, University of Duisburg, North Rhine-Westphalia, Germany). Measurements were conducted on a ULVAC-PHI VersaProbe II System using monochromatic Al Kα light with a photon energy of 1486.6 eV.

Elemental analysis was conducted by Mikrolaboratorium Kolbe (Oberhausen, Germany). The contents of silicon were analyzed after pressure digestion on an ICP-OES Model 5100 from Agilent Technologies Sales & Services GmbH & Co. KG, Waldbronn, Germany.

Field-emission scanning electron microscopy (SEM) on xerogels was performed using an Ultra 55 device from Zeiss, Germany. For the purpose of sample preparation, samples were coated with a thin conductive layer of platinum under vacuum (4 × 10^−4^ mbar) with a current of 21 mA within 90 s (SCD 500 Sputter Coater, BAL-TEC, Balzers, Liechtenstein).

Additional energy-dispersive X-ray (EDX) analyses were performed on silylated RF xerogels using an AZTec device from Oxford Instruments NanoAnalysis, High Wycombe, UK, at a voltage of 5 kV and at a working distance of 8.5 mm. EDX was used for the mapping of carbon, oxygen, and silicon.

For the determinations of the static contact angles, a water droplet (50 µL) was deposited on the (hydrophobic) solid surface of the monolith using a microsyringe (Eppendorf, Hamburg, Germany). Images were taken with a camera at a distance of 25 cm from the monolith at a 3.5× magnification and processed with the image processing software ImageJ to determine the angle between the tangent on the droplet and the monolith surface as the contact angle. Measurements were performed in triplicate (using different monoliths). The static contact angle was derived using ImageJ (version 1.52a) and applying two different fitting methods within: The first approach involved manually defining both the monolithic surface and the tangent line in order to determine the contact angle between the two lines. For comparison, a second method, based upon the low-bond axisymmetric drop shape analysis (LB-ADSA) [31] that involved fitting an ellipse around the entire droplet contour, was used in ImageJ (LB-ADSA plugin), as shown in Figure 6.

Dynamic contact angle measurements were carried out using a force tensiometer K100C from Krüss GmbH, Hamburg, Germany. The Wilhelmy plate method was applied using cylindrical gel monoliths (Figure 7). Deionized water was used as the reference liquid.

The longevity of the hydrophobic properties was evaluated by exposure to humid air (relative humidity approximately 40% at 19–20 °C) over the course of several months.

Additionally, the stability against pure liquid water, acids, and bases was assessed in analogy to ASTM 543 by submerging the monoliths in deionized water, 10% (*w*/*w*) aq. HCl solution, and 10% (m/m) NaOH solutions, respectively [34]. For this purpose, cubic samples of approx. 5 mm edge lengths were cut from the monoliths using a scalpel and submerged in the liquid. After periods of 1, 7, 14, and 52 days, respectively, the monoliths were temporarily removed from the liquids, and the soaked monoliths were weighed.

## Data Availability

The data presented in this study are openly available in the DLR repository “elib” accessible at www.elib.de (accessed on 10 April 2022).

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
