# Peer review of "Hydrophobization of Monolithic Resorcinol-Formaldehyde Xerogels by Means of Silylation"

_gels, 2022, doi:10.3390/gels8050304_

Round 1

Reviewer 1 Report

The work is very relevant and it is very well discussed. In my opinion, it deserves publication after minor changes.

In figure 4 :

a) the scale bar of the micrographs are too small and they cannot see it well .

b) the three figures should present (a) , (b) and (c) to be discussed properly in the text

c) figure 4b present two types of textures that should be commented: a kind of shell that seems to have no rugosity and a inner part with the typical nodule structure of the carbon-gels. Please give more details of the reason of these 2 structures.

In table 4.2 the surface area is extremely low (less than 1 m2/g) what is the reason for this surface area? Usually RF gels present higher surface area in the bibliography. Why is not determined the surface area in the case of TMS-B?

In tables 4.3 and 4.4 there are some values that are "not determined" and the authors do not explain the reason in the text. It should be clarified.

Figures 4.5 are too small to be distinguished properly

I cannot see the reason for the numbering of the figures and tables along the manuscript. Why all of them are 4.X? 

Reviewer 2 Report

The bond formation between xero gel and silane should also be confirmed by FTIR. 

The following article should also be incorporated in introduction part. 

 https://doi.org/10.1016/j.micromeso.2019.109697 

In section 4.1 

1-"Typical approaches involve the preparation of 1-50 gel monoliths", Please make it more elaborate have you prepared one or fifty gel monoliths?   

2-After stirring (250 rpm) at 22 °C for 5-10 min, Is it 5 minutes or ten minutes?   

3-Stirring for another 5 min resulted in a colorless solution whose pH was adjusted to 5.4- 5.6" Please make it clear that the pH of both portions was adjusted or just this portion?   

 4- ...and washed every 10-12 h with DMF, Please make it clear the time interval you used for washing and how many times you washed?   

In Figure 4. 2: SEM micrograph of ...with TBSCl (middle, right),: In the middle image, a crust is formed at the top, what is that for? 

5- Quality of the figures should be improved for clear understanding. 
